# Surface Ligand Influences the Cu Nanoclusters as a Dual Sensing Optical Probe for Localized pH Environment and Fluoride Ion

**DOI:** 10.3390/nano13030529

**Published:** 2023-01-28

**Authors:** Kumar Babu Busi, Subhalaxmi Das, Mathangi Palanivel, Krishna Kanta Ghosh, Balázs Gulyás, Parasuraman Padmanabhan, Sabyasachi Chakrabortty

**Affiliations:** 1Department of Chemistry, SRM University, Guntur 522240, Andhra Pradesh, India; 2Lee Kong Chian School of Medicine, Nanyang Technological University Singapore, 59 Nanyang Drive, Singapore 636921, Singapore; 3Cognitive Neuroimaging Centre, Nanyang Technological University, 59 Nanyang Drive, Singapore 636921, Singapore; 4Department of Clinical Neuroscience, Karolinska Institute, 17176 Stockholm, Sweden

**Keywords:** copper nanoclusters, bovine serum albumin, L-cysteine, pH sensor, fluoride ion detection

## Abstract

Functional metal nanomaterials, especially in the nanocluster (NC) size regime, with strong fluorescence, aqueous colloidal stability, and low toxicity, necessitate their application potential in biology and environmental science. Here, we successfully report a simple cost-effective method for red-/green-color-emitting protein/amino-acid-mediated Cu NCs in an aqueous medium. As-synthesized Cu NCs were characterized through UV-Vis absorption spectroscopy, fluorescence spectroscopy, time-resolved photoluminescence, dynamic light scattering, zeta potential, transmission electron microscopy and X-ray photoelectron spectroscopy. The optical properties of both Cu NCs responded linearly to the variation in pH in the neutral and alkaline ranges, and a robust pH reversible nature (between pH 7 and 11) was observed that could be extended to rapid, localized pH sensor development. However, a contrasting pH response nature between protein–Cu NCs and amino acid–Cu NCs was recorded. The alteration in protein secondary structure and strong binding nature of the surfactants were suggested to explain this behavior. Furthermore, we investigated their use as an efficient optical probe for fluoride ion detection. The limit of detection for protein–Cu NCs is 6.74 µM, whereas the limit of detection for amino acid–Cu NCs is 4.67 µM. Thus, it is anticipated that ultrasmall Cu NCs will exhibit promise in biological and environmental sensing applications.

## 1. Introduction

Functional nanomaterials in the nanocluster size regime have become increasingly ubiquitous in various commercial applications, such as bioimaging, sensors, light-emitting devices, catalysis, photosensitizers, bioelectronics and theranostics [1,2,3,4]. Due to their extremely small sizes, such nanomaterials act as a bridging link between isolated single metal atoms and bulk-sized nanoparticles and possess interesting molecule-like characteristics [5,6,7]. In particular, metal nanoclusters (NCs) have interesting physio-chemical properties, such as high fluorescence, large stokes shifts between absorbance and emission peaks, a high photostability, low photobleaching and pH sensitivity [8,9]. In particular, as compared to more scrutinized noble metal NCs, such as gold (Au) and silver (Ag), copper (Cu) NCs are relatively less explored due to their rapid surface oxidation properties. However, despite that, the scientific community is attracted to Cu NCs on account of them being inexpensive and their high abundance on Earth [10,11,12,13,14]. Furthermore, there is a vast amount of literature that suggests the achievement of their ultrasmall sizes via templated reaction synthesis protocols, in which mostly biologically relevant molecules, such as single amino acids, peptides, proteins and DNA, were incorporated as stabilizing agents [15,16]. The choice of the ligand moiety is indispensable to the final Cu NCs, as their optical properties, such as photoluminescence (PL) intensities and emission wavelengths, are largely controlled by surface ligands. These properties could be harnessed well for bio-sensing applications. Therefore, it is essential to choose the appropriate functional molecules for better surface passivation with reducing capabilities that can serve as excellent biocompatibility, improved water solubility and robust colloidal stability [17,18,19].

In recent years, there has been considerable interest in the development of techniques for the identification and detection of biologically and environmentally significant species, in the field of bio-chemical sensors [20]. Monitoring the local pH inside the cellular environment is critical, as many cellular processes are strongly pH-dependent [21]. For instance, in biological systems, pH is a key factor in the production of adenosine triphosphate (ATP) [22]. Studies surrounding the investigation of pH typically employ two major methods for measuring surrounding pH, namely, acid base and potentiometric titrations. However, they suffer from various disadvantages, such as cumbersome synthesis steps and huge experimental errors [23]. In light of this, the fabrication of a simple, straightforward and efficient detection technique, possibly capable of monitoring internal cellular pH, is pivotal. In this respect, the exploration of optical behavior as a readout of local pH may be advantageous, and fluorescent nanomaterials would be an obvious choice due to their easy identification of local pH changes via strong optical responses. Past studies have outlined methods for measuring pH in biological beings using organic fluorophores [24,25] and genetically encoded sensors (fluorescent proteins) [26,27] with fluorescence readouts. It has been suggested through these reports that they allow us to determine not only the dynamics of pH but also the absolute values, and this seems to be a useful approach. However, these methods suffer from a plethora of pitfalls, such as photobleaching and lower colloidal stability [28]. In contrast, ultrasmall fluorescent MNCs are more favorable because of their superior photostability, strong PL, high biocompatibility and minimal toxicity [29,30,31]. Even minor pH changes that affect the fluorescence intensity and spectral attributes might lead to a significant contribution to their emission characteristics. Compared to other conventional pH sensing methods, a PL-based pH sensing technique is facile and preferred due to its resilient optical response behavior. Thus, ultrasmall Cu NCs can serve as robust fluorescent reporters because they are highly stable in PBS buffer solutions of different pH ranges for biological applications.

In addition, the detection of host/guest ions inside the biological environment is important to predict the cause of any anomaly present. It is well known that fluoride (F^−^) ions contribute significantly to a variety of chemical and biological processes [32]. The typical human body contains 3 mg of fluoride, which is a vital mineral necessary for the normal development and maintenance of healthy hair, nails, teeth and bones [33]. As an example, F^−^ is frequently added to toothpaste due to its benefits to oral health and great potential in the treatment of osteoporosis. Generally, F^−^ ions are absorbed by the body easily but they demonstrate a slow-release phenomenon. As a result, long-term exposure to high levels of F^−^ ions might cause gastrointestinal issues, kidney failure, cancer and even death [34]. Moreover, the environmental protection agency (EPA) and recent research findings have revealed that the main cause of skeletal fluorosis is F^−^ ion consumption at concentrations of 1.5 ppm [35]. However, the F^−^ ion’s tiny ionic radius, high hydration energy, strong electronegativity and hard Lewis basic character make it an appealing target for separation from water, which is a significant academic challenge. To address this, the optical response technique offers an easy and fast recognition of F^−^ ions inside the aqueous environment. For such chemical sensing applications, the less-expensive and highly luminescent Cu NCs could be utilized to detect F^−^ ions, even at low concentrations such as in the µM range. 

To date, a growing body of literature has evaluated a large number of optical sensors in the detection of pH and F^−^ ions [36]. Among them, quantum dots [37] and organic dyes [33] have been reported, but they suffer from either toxicity or photostability issues. In this respect, the alteration in PL intensity in Cu NCs displays great potential in identifying and/or sensing localized pH and F^−^ ions. In addition, the reduced toxicity and biocompatible surface coating make them useful for real-time applications [38]. Herein, we have synthesized protein-templated and cysteine-coated nontoxic Cu NCs that are highly fluorescent and colloidally stable in an aqueous medium [39]. Red- and green-color-emitting Cu NCs were obtained from bovine serum albumin (BSA) and cysteine passivation, respectively. The various functionalities present in the protein backbone were identified through X-ray photo-electron spectroscopy (XPS). We have demonstrated that the PL properties of such Cu NCs are reversible depending on the local pH environment. Furthermore, the rationale underlying the contrasting PL behavior for red- and green-color NCs was illustrated. Additionally, we have illustrated that Cu NCs were capable of effectively detecting and sensing F^−^ ions in water at extremely low concentration (in the approximate µM) ranges.

## 2. Materials and Methods

### 2.1. Reagents and Chemicals

All the chemicals were purchased and utilized directly for all the experiments without further purification. Copper (II) sulphate pentahydrate (CuSO_4_·5H_2_O, ≥98.5 to 102.00%), bovine serum albumin (BSA, ≥96%) and L-cysteine (≥99%), were purchased from HIMEDIA Laboratories Pvt. Ltd. Mumbai, Maharashtra, India. Hydrogen fluoride (HF, ≥40%) and hydrogen chloride (HCl, ≥35.4%) were purchased from QUALIGENS, Thermo Fisher Scientific India Pvt. Ltd., Powai, Mumbai, India. Potassium iodide (KI), sodium chloride (NaCl, ≥99.5%) and sodium hydroxide pellets (NaOH, 97%) were purchased from FINAR Ltd. Ahmadabad, Gujarat, India. Hydrazine hydrate (N_2_H_4_·H_2_O, 80%), potassium bromide (KBr, ≥98.5%), sodium acetate trihydrate (CH_3_COONa, ≥99%), sodium hydrogen carbonate (NaHCO_3_, ≥99.5%), sodium nitrate (NaNO_3_, ≥99.5%), sodium nitrite (NaNO_2_, ≥97%), sodium phosphate monobasic (NaH_2_PO_4_, ≥99%) and acetic acid (CH_3_COOH, ≥99.5%) were purchased from Sisco research laboratories, Maharashtra, India. All experiments were carried out with Milli Q-water.

### 2.2. Methods for Fabrication

Aqua regia (HCl: HNO_3_, volume ratio 1:3) was used to clean all the glassware before each experiment to minimize contamination. Following that, they were rinsed with deionized water and dried in a drying oven. The synthesis procedure of red-color-emitting BSA-derived Cu NCs and L-cysteine-mediated bright-green luminescent Cu NCs is described in the below section.

#### 2.2.1. Synthesis of Red-Emitting BSA-Cu NCs

Busi et al. previously reported the synthesis of BSA-Cu NCs in aqueous medium [39]. Briefly, 3 mL of concentrated CuSO_4_·5H_2_O at 5 mM was mixed with 150 mg of BSA after the reaction medium reached 55 °C. After 10 min, NaOH (1 M) solution was added to the reaction to bring the pH to 9, which was shown by the solution gradually changing from milky white to violet in color. The second reducing agent, N_2_H_4_·H_2_O, was added to the reaction mixture after 10 min to reduce the Cu^2+^ oxidation state to the Cu^0^ state. The reaction was then allowed to proceed for 75 min at 55 °C with continual stirring to result in colloidally stable, highly fluorescent Cu NCs before purification. Finally, the reaction mixture was stored at 4 °C for subsequent advanced characterization. 

#### 2.2.2. Synthesis of Green-Emitting L-Cysteine-Cu NCs

Through a slight modification to the reaction protocol, as described by Soumyadip et al., we created bright-green luminescent Cu NCs that are mediated by a single amino acid [40]. Initially, 19 mL of 0.1 M CuSO_4_.5H_2_O was mixed with 5 mg of L-cysteine. After 10 min, NaOH (1 M) solution was introduced to the reaction to raise the pH to 12. Finally, the reaction mixture was allowed to react for 5 h at 37 °C. For further characterization, the reaction mixture was then stored at 4 °C.

#### 2.2.3. Preparation of Buffer Solution

The acidic buffer solution (pH 5) was made with 0.1 M concentrations of CH_3_COOH and CH_3_COONa, while the neutral and basic buffer solutions (pH-7, pH-11) were made with 0.1 M concentrations of NaH_2_PO_4_, HCl and NaOH. To adjust the pH of solution we used a pH meter from Eutech Instruments Pte Ltd, Singapore 139949.

#### 2.2.4. Detection of Fluoride Ion

The fluoride ion (F^─^) concentration was determined by the fluorescence intensity of Cu NCs. Cu NCs were functionalized with two different surface capping groups, namely, BSA and L-cysteine; the PL intensity was recorded at 650 nm and 492 nm, respectively, whereby the excitation wavelength was set to 365 nm for both the NCs. The limit of detection (LOD) value was evaluated based on the calibration data as 3 σ/k, where k is the slope of the calibration curve and σ is the standard deviation from the blank measurements (the plot of concentration of F^─^ ion vs. quenched fluorescence intensity) [36]. 

### 2.3. Measurements and Characterizations

*UV-Visible spectroscopy:* To measure absorbance and PL, Greiner-96-well plates with a 200 µL capacity were employed. The microplate reader used was TECAN Spark M, and it was operated in both the absorbance and fluorescence intensity scan modes using XENON lamp source. 

*Transmission Electron Microscope:* A high-resolution transmission electron microscope (JEOL-JEM 2100) was used to take bright-field TEM images at a high voltage of 200 kV. The copper grid (200-mesh, from TED PELLA, Inc. Redding, CA, USA) was drop-casted with 5–10 µL of BSA–Cu NCs and allowed to dry overnight under incandescent bulb (60 W) to eliminate the moisture on the copper grid. All the TEM images were analyzed with the help of ImageJ software. 

*Time-Resolved Photoluminescence (TRPL):* Time-correlated single-photon counting (TCSPC—Horiba Jobin Yvon IBH) spectrometry was employed to obtain TRPL measurements. At repetition rates up to 100 MHz, the laser diode (Delta Diode—425L) may produce light pulses with a standard width of 100 ps and a peak power of 230 mW. The excitation laser source peak wavelength was 420 nm, and the measurement range for sample lifetime was 6500 ns. The PL decay curves were further examined using IBH DAS6 software.

*X-ray Photo-electron Spectroscopy (XPS):* For the XPS measurements, Mg Kα (1253.6 eV) radiation was used to determine the surface elemental analysis of the sample (PHI VersaProbe III). The measurement was performed at a 45 °C detection angle, employing pass energies of 55 eV for survey spectra and 280 eV for detailed spectra by the analyzer, respectively. The sample’s precise surface spot size (5 × 5 mm) was chosen for the XPS investigation. In order to counteract the surface’s charge effects, the samples were neutralized using electrons from a flood gun (20 µA current). The protein-mediated Cu NC sample was sputter-cleaned. The instrument’s energy resolution was ≤0.5 eV, which was applied to calibrate the binding energy for the Cu peaks used at 932.6 eV. 

*Dynamic Light Scattering (DLS) and Zeta Potential:* Malvern Zetasizer Nano ZS was used to measure the Cu NCs ξ-potential in electrical double-layered cells and hydrodynamic size in a disposable sizing cuvette at 25 °C.

*Circular dichroism:* Circular dichroism (CD) measurements were taken on the AVIV Model 420 Circular Dichroism Spectrometer (Tel Aviv University, Tel Aviv-Yafo, Israel). Quartz SUPRASIL precision cells of path length 0.01 mm (capacity 25 μL) were used for all the measurements (Hellma Analytics, Müllheim, Germany). The measurements were taken between a range of 180 and 260 nm at 25 °C. All readings were taken in duplicates, before being averaged to produce the final spectra. A concentration of 1 mg/mL was used for all samples. 

## 3. Results and Discussions

### 3.1. Design of BSA-Cu NCs as Biosensor

Aqueous solutions of red-color-emitting, high-fluorescent BSA-Cu NCs were obtained by following our previously reported protocol [39]. These BSA-Cu NCs were colloidally stable in the long run. These NCs were extremely small in dimension and their optical and structural characteristics are presented in Figure 1. The successful synthesis of these Cu NCs was confirmed through the absorbance spectra, where no observable peak was obtained. In addition, the suppression of the characteristic 540 nm surface plasmon resonance (SPR) peak for larger Cu nanoparticles (NPs) validated the non-existence of such larger sizes. These BSA-Cu NCs showed a strong excitation peak at 367 nm and the highest emission intensity at 652 nm, both accompanied by a large Stokes shift at around 285 nm, as shown in Figure 1a. These as-prepared BSA-Cu NCs were able exhibit an excitation-independent stable emission in the red region 652 nm, as depicted in Appendix A. This behavior was of the utmost importance for Cu NCs to be used as rapid sensors, so as to avoid crosstalk during selective sensing applications in the complex environment. In order to comprehend the excited-state relaxation process, we additionally examined the radiative decay dynamics in Figure 1b. The radiative decay curve was tri-exponentially fitted and the components were 0.18 (7.43%), 2.01 (88.19%) and 0.002 (4.38%) µS, and, according to time-correlated single-photon counting (TCSPC) observations, the average lifetime of Cu NCs was 1.78 µS. The singlet–triplet transition is believed to be involved in the triexponential decay, which confirmed that the emission originated from three different emissive states. The as-synthesized Cu NCs are represented in a digital image in Figure 1c, whereby a yellowish color in ambient light and intense red color under UV light (365 nm) illuminations were observed. The Cu NCs possessed a negative ξ-potential (~−25.23 mV), which may be attributed to the deprotonation of the numerous -COOH groups on the protein residues. Moreover, the high ξ-potential value of Cu NCs demonstrated that these NCs were remarkably stable in an aqueous medium. For a better understanding about their dimensions, we investigated their size distribution, morphology and crystalline structure through advanced HRTEM characterization. The bright-field TEM image (Figure 1d) of Cu NCs clearly indicates the formation of well-embedded spherical-shaped NCs associated with the protein, with an average diameter of 2.27 ± 0.33 nm. The histogram generated by taking into consideration 40 representative small Cu NCs in TEM images is depicted (*inset* in Figure 1d), in which the spherical NCs are marked by yellow-colored dotted circles. Figure 1e represents the crystalline nature of Cu NCs and the lattice spacing measurement of 0.205 nm confirmed the existence of a Cu (111) plane in their structure. The selected area electron diffraction (SAED) pattern (Figure 1f) suggested that the as-synthesized Cu NCs exhibited a finite polycrystalline nature. Additionally, we used the DLS measurement to quantify the hydrodynamic radius, which revealed an average size of 6.48 ± 0.07 nm, which was in accordance with the size of BSA. We believe that the Cu NCs were completely embedded inside the protein matrix/backbone. 

Among the three coinage noble metals, Cu is easily prone to oxidation due to its low reduction potential [19]. Consequently, it was deemed essential to determine the oxidation state of Cu during cluster preparation to determine its stability. In order to realize the surface elemental composition of protein-mediated Cu NCs, high-resolution XPS analysis utilizing Al Kα (hν—1486.6 eV) radiation was performed. The full survey scan spectrum in Figure 2a confirms the existence of various elements, such as C, N, O and S, which were observed with the protein backbone present with Cu NCs [41]. As depicted in Figure 2b, two separate broad peaks were identified in the Cu 2p spectra at binding energies of 931.9 eV and 951.6 eV, which can be attributed to Cu 2p_3/2_ and Cu 2p_1/2_, respectively. Additionally, a minor shakeup at 942 eV signified that the NCs apparently include a Cu (II) oxidation state [42] as well. However, it was not possible to distinguish between oxidation states 0 and +1 due to their highly similar 2p_3/2_ binding energies. Therefore, we concluded that inside the protein matrix, the Cu atoms most likely stayed in a mixed valence state [43]. Subsequently, the C1s peak investigation of the BSA-Cu NCs disclosed two different peaks in Figure 2c, which relates to the binding energies of C=C/C-C (283.48 eV) and C-C/COOH/C-N (285.32 eV) present in the protein backbone, respectively. Similarly, the N1s element showed two peaks due to the presence of NH_3_ (398.33 eV) and CH_3_-C-N (397.07 eV) in Figure 2d, whereas the O1s element displayed two peaks in Figure 2e, attributed to the O**=C-OH (530.43 eV) and O=C-O**H (533.63 eV), respectively. Interestingly, Figure 2f suggests that the availability of the Cu (I) oxidation state on the cluster’s core surface was consistent with the presence of an intermediate species. The S2p element provided two peaks that were attributed to oxidized sulfur (SO_2_—167.88 eV) and sulfur copper (Cu-S—162.14 eV). The strong thiol and metal bonds provide excellent colloidal stability to the cluster core and protect them from the aggregation inside the aqueous medium [44]. Therefore, the four components demonstrated the presence of a protein backbone in the produced Cu NCs.

### 3.2. Effect of pH on the Fluorescence Properties of BSA-Cu NCs

In general, pH is a critical parameter for understanding and facilitating numerous chemical reactions, and physiological activities of cells and organelles. As a result, reliable determination of localized pH within the body is important in the field of chemical biology. Interestingly, noble metal nanoclusters responded well to pH [45] by virtue of their pH-dependent fluorescent behavior. Here, we explored the effect of pH on the fluorescence emission property of BSA-Cu NCs. The emission efficiency was evaluated in the presence and absence of three different universal phosphate buffer (5, 7 and 11) solutions (PBS) with 0.1 M concentration and their outcomes are presented in Figure 3. As depicted in Figure 3a, no discernible alteration in the absorbance peaks of BSA-Cu NCs was observed after the addition of different buffer solutions, such as 5, 7 and 11. Further, we did not observe any aggregation in those solutions. Figure 3b shows that upon the addition of the acidic buffer pH 5 and pH 7, there was a slight decrease in PL intensity. However, excellent colloidal stability of the BSA-Cu NCs was maintained in those buffer media. In contrast, after adding pH 11, we noted a significant reduction in fluorescence intensity that might have arisen due to the intermolecular interactions between the surface-capping protein and the guest/host ions in the buffer solution. In addition, we conducted TRPL measurements to those BSA-Cu NCs after the addition of buffer solutions, shown in Figure 3c. As a result, the relative amplitude of the radiative lifetime decay patterns (the dominant contributor) was significantly decreased with pH 5 (85.48%), pH 7 (84.18%) and pH 11 (79.76%) buffers, as compared to the control measurement using Milli-Q water (90.35%). Here, during the intra-band transition between the highest occupied molecular orbital and lowest unoccupied molecular orbital, the contribution from non-radiative transition related to the presence of defects and increases (τ1) is measured, shown in Appendix A [46]. Furthermore, the zeta-potential studies demonstrate that the Cu NCs were optimally colloidally stable in PBS buffer solutions at three different pH ranges. However, the Cu NCs surface charge was altered with the addition of PBS buffer solution, as compared to the control measurement, due to the presence of various ions. As the pH was lowered, it is possible that more secondary structures and free random coil structures occurred in the protein matrix, consequently increasing the surface charge (positive side), as shown in Appendix A. Therefore, the BSA-Cu NCs optical attributes responded extraordinarily well to PBS buffer solutions of different pH ranges and may be better suited for detecting the changes in pH dynamics under various medical conditions.

### 3.3. Cu NCs as Versatile pH Sensor

In addition to their reversible nature, their attractive PL attributes would certainly add value to the robustness in using BSA-Cu NCs as pH sensors. Interestingly, we observed an excellent pH reversibility in a pH range of 7 to 11, where most of the biological entity operates. In Appendix A, the procedure for calculating pH reversibility is described in depth. We clearly observed the modifications in the PL properties of the BSA-Cu NCs by adding neutral (pH-7) and alkaline (pH-11) in Figure 4a. These changes in the PL of BSA-Cu NCs at various pH levels could be broadly ascribed to the corresponding structural modifications in the BSA protein [47]. It is notable that a very strong fluorescence enhancement without any emission shift or intensity loss was observed in the BSA-Cu NCs at a neutral pH, which might indicate significant variations in structural binding changes to amino acid side chains. To interpret the structural variations in the protein, we performed the circular dichroism (CD) spectral measurements at various pH levels, which are shown in Figure 4b. When the pH is lowered, the α-sheet secondary structure of BSA deformed and transformed into a β-sheet structure, predominantly accompanied by flexible random coil structures. This phenomenon is believed to have occurred due to structural rearrangements at a high pH, caused by titration of side chains of amino acid residues, which consequently increased the surface passivation on the NCs. Additionally, an increase in fluorescence was observed as a consequence of the increasing availability of functional groups at neutral pH, including -OH, -NH_2_ and -COOH. On the contrary, at an alkaline pH (pH 11), the fluorescence intensity steadily decreased, and we hypothesized that the reduced availability of functional groups owing to the decreased surface passivation towards NCs might have been responsible for this phenomenon. Thus, the observed trends in the optical properties of BSA-Cu NCs may be significantly influenced by both the dramatic shifts in secondary structure of the protein and inherent nature of Cu NCs. 

To understand this better, we synthesized single amino acid L-cysteine-mediated Cu NCs (Cys-Cu NCs) in an aqueous solution to further realize this pH-responsive behavior of NCs. The synthetic process, which used a straightforward chemical-reduction technique, is described in more detail in Section 2.2.2. The optical characteristics of the as-prepared Cys-Cu NCs are described in Appendix A. The high-resolution TEM image reveals an average size ~1.98 ± 0.2 nm, as depicted in Appendix A**.** Surprisingly, we discovered that the Cys-Cu NCs displayed opposing pH-reversible behavior in a pH range of pH 7 to 11, as compared to BSA-Cu NCs. For Cys-Cu NCs, the absorbance and PL spectra were collected in the presence of three different buffer solutions, which are included in Appendix A**,** and we followed the same pH-reversibility strategy as previously detailed. We observed that Cys-Cu NCs produced strong emission spectra at alkaline pH 11, whereas the intensity of the emission is reduced at neutral pH 7, as depicted in Figure 4c. We hypothesized that the deprotonation of the thiol groups (-SH^−^ to -S^2−^) occurs at an alkaline pH (pH 11), which enabled more interaction with the surface of Cu NCs for better passivation [40]. Interestingly, the zeta-potential measurement also disclosed that the surface charge increases with the addition of alkaline pH 11 and is almost similar to control measurements, as shown in Appendix A. However, the effective surface passivation was also high due to strong thiol bonding at an alkaline pH, significantly enhancing the fluorescence intensity. In addition, cysteine may serve as a surface protector and a reducing agent simultaneously, without the necessity for an external reducing agent. Therefore, the surface functionalization of NCs plays an important role in pH sensing applications because the selectivity and sensitivity properties may be altered as per the surface-capping groups present. Furthermore, we performed a PL stability study with the addition of other anions to the as-prepared two-ligand-mediated Cu NCs.

### 3.4. Anion Selectivity Test

The selectivity of various cations/anions and further sensing of those inside the same environment is important because they play a decisive role in a wide range of chemical and biological processess. Further, the change in the optical response of Cu NCs due to the interaction of guest ions might be useful for further applications. Among them, sensing and recognition of the F^−^ ion has become a topic of interest, due to its relevance in both chemical and biologcal processess, as mentioned earlier. To investigate that, we executed an anion selectivity test with both the colour-emitting Cu NCs against other typical anions in the presence of their corresponding cations. To determine their selectivity, we chose H^+^(F^−^), Na^+^(Cl^−^), K^+^(Br^−^), K^+^(I^−^), Na^+^(HCO_3_^−^), Na^+^(H_2_PO_4_^−^), Na^+^(NO_3_^−^) and Na^+^(NO_2_^−^), at a concentration of 0.1 M each. These anions were specifically chosen due to their presence in drinking water. As depicted in Figure 5a,b, the emission intensity did not change greatly in the presence of other anions, but the fluorescence decreases significantly with the addition of F^−^ ion at a 0.1 M concentration. In addition, we performed the TRPL measurements for both ligand-mediated Cu NCs, and the non-radiative decay patterns were enhanced after the addition of the F^−^ ion. However, the lifetime of other anions, such as Br^−^ and NO_3_^−^, was almost similar to that of the control measurements, as shown in Figure 5c,d. These findings suggest that the two different colour-emitting ultrasmall Cu NCs may detect F^−^ ions in aqueous media.

### 3.5. Fluoride Ion Detection of Cu NCs

The calibration curves, corresponding to both BSA-Cu NCs and Cys-Cu NCs, were plotted using the F^−^ ion as a quenching agent at different concentrations. As recorded in Figure 6, the fluorescence performance of the two different color-emitting Cu NCs can be utilized as an efficient fluorescent probe for F^−^ ion detection. F^−^ ion concentrations ranging from 0.1 to 100 µM were taken for BSA-Cu NCs, which were purposefully chosen because of their high sensitivity, even at low concentrations. The subsequent effect on PL intensities is presented in Figure 6a. We observed a gradual quenching in fluorescence intensity with an increasing concentration of F^−^ ions. Figure 6b shows that a linear (R^2^ = 0.9816 for BSA-Cu NCs) relationship between F^−^ ion concentration and fluorescence intensity could be obtained. The calculated limit of detection (LOD) value was 6.74 µM, which was similar to the lowest possible detection limit present in related NC literature [48,49]. Similarly, for Cys-Cu NCs, as shown in Figure 6c, the concentrations of F^−^ ions were measured from 0.1 to 100 µM and similar trends were recorded. Figure 6d depicts that the optical characteristics of Cys-Cu NCs were found to be linearly correlated (R^2^ = 0.9911) with the F^−^ ion concentration, and the LOD was calculated to be 4.67 µM. In addition, the parameters for the detection limit measurements for both red- and green-color-emitting Cu NCs are provided in Appendix A. Therefore, irrespective of the ligand, the ultrasmall Cu NCs displayed excellent sensitivity towards F^−^ ion, even at the lowest concentrations. Finally, we measured the F^─^ ion concentrations in real-time samples of tap water, lake water and river water that were spiked with a 10 µM concentration, as shown in Appendix A. These findings confirm that both ligand-mediated Cu NCs can detect F^─^ ions in water samples. Therefore, due to the excellent optical response, these NCs could be employed as potential optical sensors in biological and environmental applications.

## 4. Conclusions

Surface ligands had a prior importance in protecting the metal core and altering their optical response. In this study, we effectively synthesized and characterized two different ligand-functionalized Cu NCs in aqueous media, with distinct color-emission characteristics. Interestingly, we noticed that the surface functionalization had a significant influence on the PL properties of both Cu NCs while they were incubated with different pH solutions. They were colloidally stable in those solutions, and both showed reversible pH-responsive behavior. However, their responsive nature was not the same. Based on their ligand dynamics, we explained that the change in secondary structure of the BSA protein and binding affinity of S^2−^ played a decisive role. In addition, we discovered that these Cu NCs could also be employed as an excellent optical probe for fluoride ion detection while treating them with various other anions. We observed that this fluoride ion detection was independent of the surface ligand on Cu NCs. Here, the BSA-Cu NCs had a detection limit of 6.74 µM, while Cys-Cu NCs showed a detection limit of 4.74 µM. Thus, these ultrasmall fluorescent Cu NCs might be utilized as an efficient optical probe for detecting the dynamics of localized pH and fluoride ion in real-time applications. This research demonstrated the potential to open new avenues for the detection of small molecules or local environment inside biological systems through PL responsiveness, especially for in vivo applications.

## Figures and Tables

**Figure 1 nanomaterials-13-00529-f001:**
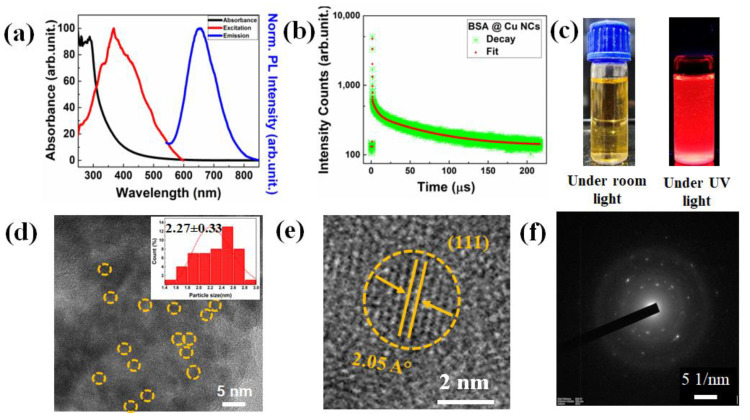
(**a**) Optical characteristics of BSA-Cu NCs, including absorbance, excitation and emission spectra (black, red and blue color line). (**b**) Time-resolved PL radiative decay patterns of Cu NCs, showing a three-exponential decay. (**c**) Digital photographs were captured while these Cu NC solutions were being exposed to the room light and under 365 nm UV light irradiation. (**d**) The inset histogram analysis and yellow-colored short, dotted lines on the low-resolution HRTEM image of Cu NCs indicates that the average size is ~2.27 ± 0.33 nm. (**e**) The high-resolution image of BSA-Cu NCs revealed the Cu (111) plane with 2.05 A° of d-spacing. (**f**) The SAED patterns confirmed the low crystalline nature of Cu NCs due to their ultrasmall size.

**Figure 2 nanomaterials-13-00529-f002:**
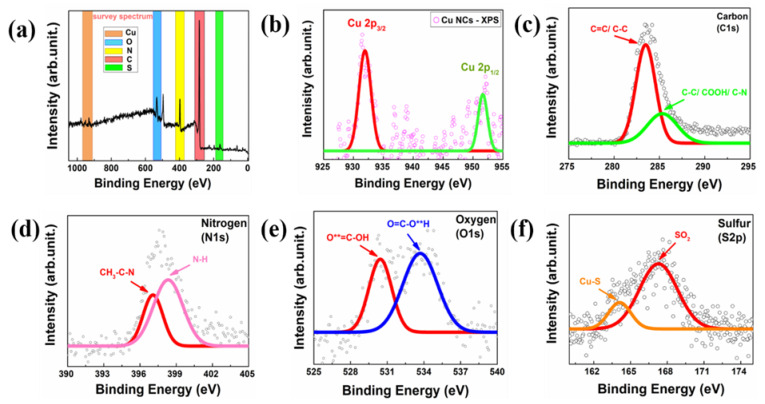
The XPS analysis of BSA-Cu NCs. (**a**) The existence of C, N, O, S and Cu elements in the protein-mediated Cu NCs was confirmed by the full survey scan. (**b**) High-resolution XPS revealed the presence of Cu 2p_3/2_ and 2p_1/2_ binding energies ascribed to the zero (red line) and monovalent (green line) oxidation states of Cu NCs. (**c**) C1s element shows two distinct peaks, which represent C=C or C-C (red line) and C-C/COOH/ C-N (green line), respectively. (**d**) N1s element showed two peaks, CH_3_CN (red line) and NH_3_ (pink line). (**e**) O1s element confirms the presence of O**=C-OH (red line) and O=C-O**H (blue line). (**f**) S2p element observed multiple peaks corresponding to sulfur copper (Cu-S) (yellow line) and oxidized sulfur (SO_2_) (red line). The NIST-XPS database was utilized to validate the bonds for different BE values, and the black dots in all the pictures (**b**–**f**) represent the experimental results.

**Figure 3 nanomaterials-13-00529-f003:**
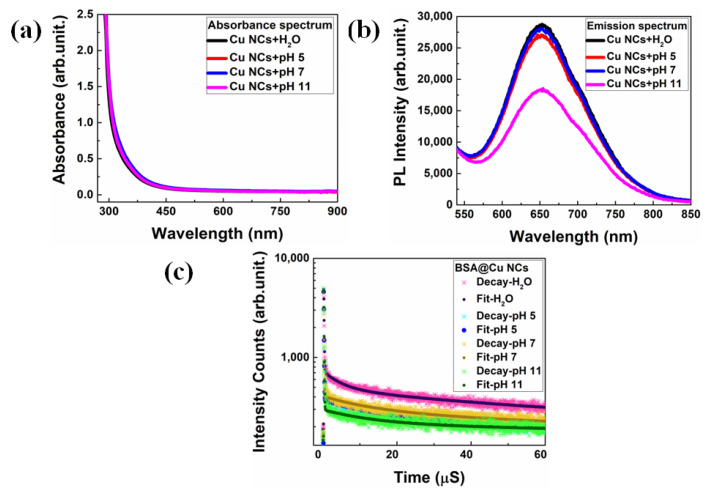
The effect of buffer solutions (pH 5, 7, 11) on the optical properties of BSA-Cu NCs. (**a**) No sharp SPR peak observed after the addition of buffers to the BSA-Cu NCs. (**b**) The emission intensity gradually decreased due to the presence of pH 7 buffer solution to the NCs. (**c**) The radiative decay patterns also followed the decrement trend in the NCs after incorporating different buffers.

**Figure 4 nanomaterials-13-00529-f004:**
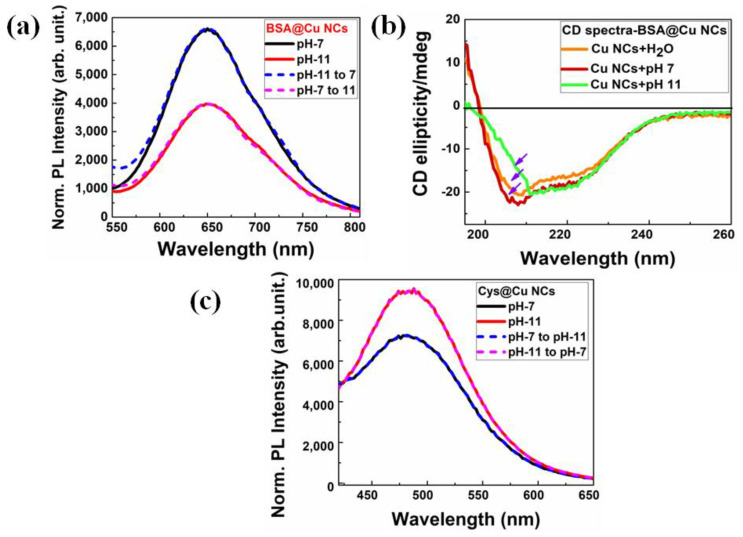
Measurement of pH reversibility on the optical characteristics of two different color-emitting Cu NCs. (**a**) Due to structural alterations in the protein, BSA-Cu NCs exhibited high emission behavior at neutral pH and weak emission intensity at pH 11. (**b**) CD spectra of BSA-Cu NCs revealed the secondary structural changes in BSA protein. The purple arrows indicate the structural changes in the protein. (**c**) Due to the availability of more thiol groups at higher pH 11, where emission decreased at the lower pH 7, single-amino-acid cysteine-mediated Cu NCs displayed unique behavior when compared to protein NCs.

**Figure 5 nanomaterials-13-00529-f005:**
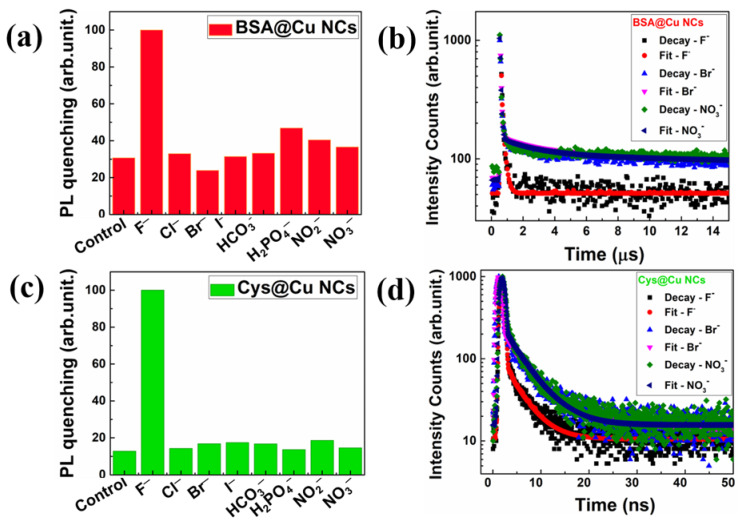
The change in the fluorescence intensity and the radiative decay patterns were measured with two different color-emitting Cu NCs in the presence of various anions at 0.1 M concentrations. (**a**,**b**) BSA-Cu NCs and (**c**,**d**) Cys-Cu NCs.

**Figure 6 nanomaterials-13-00529-f006:**
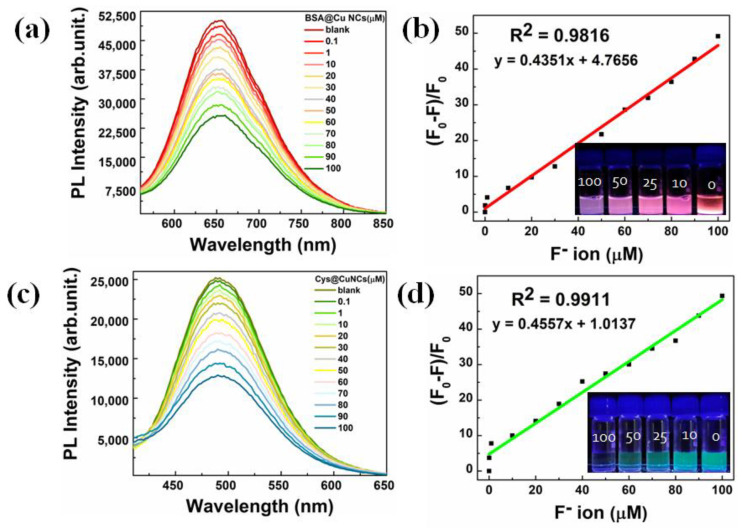
Optical characteristics of ultrasmall Cu NCs were evaluated with presence and absence of quenching constant, whereas the concentration of the fluoride ions was in s range of 0.1–100 μM. (**a**,**b**) BSA-Cu NCs and the linear curve (inlet shows the digital representation at various concentrations of F^−^ ion). (**c**,**d**) Cys-Cu NCs and the linear curve (inlet shows the digital representation at various concentrations of F^−^ ion).

## Data Availability

The data is available from the corresponding author(s) on reasonable request.

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
