# Peer review of "Surface Ligand Influences the Cu Nanoclusters as a Dual Sensing Optical Probe for Localized pH Environment and Fluoride Ion"

_nanomaterials, 2023, doi:10.3390/nano13030529_

Round 1

Reviewer 1 Report

The paper presents interesting findings. However, the work should be revised.

Introduction, Page 3, line 61. The meaning of the sentence is not clear, probably a verb is missing.

I would like to point out that the methods for measuring pH in living objects using organic fluorophores and genetically encoded sensors (fluorescent proteins) with fluorescence readout have now been developed. They allow to determine not only the dynamics of pH, but also the absolute values. This information should be included in the introduction.

It is not clear how NCs are principally different from other probes in terms of signal measurement. (Line 71).

Page 4 line 90. What is “minute concentrations”?

Line 98 – toxicity was not estimated in the manuscript.

Page 118 – milli Q or double distilled water?

Page 123 – remove word “clearly”. Previous “were” is incorrect. This is just examples; language editing is necessary for the manuscript.

Page 6, line 197 – is “recombination” indeed the correct term?

Line 204 – information about pI of BSA will clarify the ξ-potential.

Line 205 – “protein” or “protein residues” not “protein backbone”.

Figures 1, 3-5. Concentration of NCs and buffers content should be indicated in text, figures captions or Materials and Methods section.

Page 8, line 274 – which concentration of phosphate was used?

Phosphate is not good for pH 11. It is pH of Na2HPO4, but it is far from its pKa2 and pKa3.

Line 298 – Intensiometric readout of fluorescence could not be used to measure pH, it could be used only for monitoring of dynamics of pH. Only ratiometric probes allow to establish pH value. So, presented NCs are not “suitable for detecting biological pH”.

Besides, authors do not provide pKa of pH transition. To do this, fluorescence measurement at accurately measured pHs should be done. Appropriate buffers for corresponding pH ranges should be used.

Line 285 – it is not clear, amplitudes of which “decay patterns” were established. % relative to what? Please, specify in the main text.

Page 9, line 321 – “broken hydrogen bonds” is not main case. Old bonds disappear, new arise. Structural rearrangements at high pH are due to titration of side chains of amino acid residues.

Page 10, lines 354, 355 – high emission or weak emission? Please, rephrase the sentence.

Page 11, line 368 – why HF was used instead of NaF? In which form was it used? Was pH monitored in case of F- detection? Please, clarify.

Lines 369, 372, 380 – concentrations differ (0.1 / 10 mM).

Line 393. How limit of detection (LOD) was established? Please specify in Materials and Methods. Simply, LOD is SD of signal noise multiplied by 3.3. From the presented graphs (figure 6b,d), one can see that deviations from line is in uM range, and LOD is about 10 uM. Please, read literature about correct LOD determination (ideally, you also should measure deviations of signal near and below LOD).

Figure 6b,d – vertical axis indeed is not (I-I0)/I0. At concentration 0 it should be 0. Clarify, what is I0.

Line 421 – Cu NCs could not be used “for the detection of pH” (value?), only dynamics of pH can be measured. And contrast is not very high.

Page 13, line 423 – no evidence of applicability of the obtained NCs in vivo were presented. I believe that it is preliminary to talk about applicability of the obtained NCs in living organisms.

Please, save the Supporting Information file as PDF since it is more universal.

Figure S1 – Spectra of “Excitation independent emission” are indeed quite different. Probably, title is not correct.

No data regarding size of Cys-Cu NCs were presented in the manuscript. It could be interesting to compere size of both NCs in light of “Cu NCs embedded inside the protein matrix” (Page 6, line 219).

Author Response

Thank you for the comments and feedback on our paper. We have herewith attached our reply to reviewer 1's comments.

Reviewer 2 Report

The authors of the manuscript "Surface Ligand influences the Cu Nanoclusters as a Dual Sensing Optical Probe for localized pH-environment and fluoride ion" describe  the synthesis and characterisation of Cu nanoclusters. Their optical properties could be used as a pH-sensor, but protein and amino acid binding influences the system.
In addition, the authors investigated the use of the Cu nanoclusters as a fluoride-ion detection probe.

The manuscript needs improved grammar, and units should be consistently used (i.e., 5 nM, instead of 5nM, etc.)
Overal, the manuscript is interesting, although the claimed in vivo application as a pH sensor is not substantiated.
How, for example, could the Cu-nanoclustres be imported into living cells, taken into account the toxicity of Cu?
The use as a F--ion sensor is interesting, but should be tested in i.e. polluted water, to see how contaminants influence the results.

Author Response

Thank you for the comments and feedback on our paper. We have herewith attached our reply to reviewer 2's comments.

Round 2

Reviewer 1 Report

Authors tried to answer all comments, but some answers could not be accepted. They present their own thinking about “colloidal stability”, “biocompatibility and toxicity” of “organic fluorophores and genetically encoded sensors” in comparison to “ultrasmall fluorescent MNCs”. But no new references were added. So, authors declarations seem to be wrong, and they not proved by literature data. All statements at least in the introduction should be proved by the literature.

 Line 96 – change “minute concentrations” to “low concentrations”

Please, add reference to the method of LOD estimation that you used (“We have followed the traditionally used method to measure the LOD value for Cu NCs “Dalton Trans., 2016,45, 811-819””). As well as add references to the new data in the introduction.

As it is clearly seen from Figure 6, the standard (SD) deviation of the experimental points from the approximation line does not allow to get LOD < 1 uM. Please provide in the answer to my comments, equations of linear regressions and standard errors of the obtained parameters.

And you did not measure statistically significant data in the concentrations below 1 uM to estimate SD. But this can improve LOD and make you estimations more correct.

Besides, it is seen that the decrease of spectra intensity is almost linear in the tested region of concentrations (Fig. 6a,c). So, F0/F should be a hyperbola (Fig. 6b,d). Please present the true data. Why some points disappeared in the new version of the manuscript?

 English is still should be revised (for example, line 349 “decreased surface passivation towards NCs _could have been_ responsible for this phenomenon”).
